# Fenton Reaction–Unique but Still Mysterious

Frantisek Kastanek [1], Marketa Spacilova [1,*], Pavel Krystynik [2], Martina Dlaskova [1] and Olga Solcova [1]

[1] Institute of Chemical Process Fundamentals of the Czech Academy of Sciences, Rozvojova 1/135, 165 00 Prague, Czech Republic

[2] Faculty of Environment, University of J. E. Purkyne in Usti nad Labem, Pasteurova 3632/15, 400 96 Usti nad Labem, Czech Republic

\* Correspondence: spacilova.marketa@icpf.cas.cz; Tel.: +420-220-390-280

**Abstract:** This study is devoted to the Fenton reaction, which, despite hundreds of reports in a number of scientific journals, provides opportunities for further investigation of its use as a method of advanced oxidation of organic macro- and micropollutants in its diverse variations and hybrid systems. It transpires that, for example, the choice of the concentrations and ratios of basic chemical substances, i.e., hydrogen peroxide and catalysts based on the $Fe^{2+}$ ion or other transition metals in homogeneous and heterogeneous arrangements for reactions with various pollutants, is for now the result of the experimental determination of rather randomly selected quantities, requiring further optimizations. The research to date also shows the indispensability of the Fenton reaction related to environmental issues, as it represents the pillar of all advanced oxidation processes, regarding the idea of oxidative hydroxide radicals. This study tries to summarize not only the current knowledge of the Fenton process and identify its advantages, but also the problems that need to be solved. Based on these findings, we identified the necessary steps affecting its further development that need to be resolved and should be the focus of further research related to the Fenton process.

**Keywords:** Fenton reaction; Fenton-like reactions; advanced oxidation processes; wastewater treatment

## 1. Introduction

The Fenton reaction, discovered at the beginning of the 20th century [1], is primarily based on the idea of the formation of oxidizing radicals, which are created by the catalytic action of $Fe^{2+}$ on the decomposition of $H_2O_2$, added in a certain amount and ratio to water, which also contains various organic substances. Oxidative radicals oxidize the present organic substances to varying degrees during the momentary period of their existence. This Fenton reaction is homogeneous because the catalyst ($Fe^{2+}$ ion) is dissolved in water. However, the catalyst can also be heterogeneous. For example, the Fenton reaction is heterogeneous in the case of Fe particles, which under certain conditions release Fe ions into the solution.

The homogeneous Fenton reaction is highly popular, for instance, due to the decontamination of various organic pollutants in wastewater, both municipal and industrial. This reaction is capable of decomposing even more structurally complex organic substances, e.g., azo dyes orange II, tartrazine, nonsteroid antiphlogistics such as acetylsalicylic acid and most pharmaceuticals, as well as bacteriostatic antibiotics, e.g., tetracycline, 2,4-dichlorophene (precursor of the herbicide 2,4-dichlorophenoxyacetic acid), bisphenol A, various endocrine disruptors, e.g., estrogens, pesticides of all types, etc. On the contrary, it shows lower efficiency for the decomposition of simpler chlorinated substances, e.g., tri- and tetrachlorinated ethanes, chloroform, or tetrachloromethane [2]. The process of contaminant oxidation in an aqueous environment using the homogeneous Fenton reaction consists of several successive steps, namely the addition of reagents $Fe^{2+}$ and $H_2O_2$ in appropriate concentrations and ratios to the contaminated water while adjusting the pH,

usually to pH 3, the progress and control of the oxidation reaction, the alkalization of the solution after the reaction, and the coagulation–flocculation of the sludge and its separation.

The Fenton reaction is a pillar of the so-called advanced oxidation processes, which currently belong to the most important categories of chemical processes aimed at removing hazardous substances from the environment. To elucidate, a few selected interesting publications related to this reaction [3] are listed below. They are focused on the Fenton reaction in real wastewater from various industrial productions [4,5], wastewater from textile dyeing [6], leachate from landfills [7], fire extinguishing water containing phenol [8], and the removal of extracellular polymers in sewage sludge worsening its drainage [9]. There are already hundreds of publications dealing with the Fenton reaction, and its popularity continues even nowadays, whereby the attention of scientists addressing ecological issues is turning to previously poorly observed environmental pollutants, the so-called emerging pollutants. These previously unmonitored substances, contained in waters in concentrations of ng/L to µg/L, such as the so-called endocrine disruptors (hormones, surfactants, additives to plastics and cosmetics, phthalates, parabens, bisphenol A, nonylphenol) [10–12] or waste pharmaceuticals and drugs [13] are unwanted waste products of anthropogenic origin that negatively affect aquatic diversity and are also potentially dangerous for humans. This study summarizes the current knowledge of the Fenton process and identifies its advantages and also the problems that need to be solved, considering its historical role. Based on these findings, the necessary steps are mentioned, which can affect its further development and should be the focus of further research related to the Fenton process.

## 2. Advantages of the Fenton Process and Its Historical Role

The Fenton process is based on inexpensive reagents and the reactor arrangement is simple. Compared to other oxidation systems, e.g., gas–liquid (chlorination, ozonation), the reaction is not negatively affected by mass transfer. Concurrently, the system of Fenton components can also be bubbled with air or oxygen, similar to the heterogeneous Fenton reaction, where dissolved oxygen in water accelerates the formation of $Fe^{2+}$ in the case of using solid Fe particles as their source. The reaction can be easily controlled, and the Fenton process is indispensable if simple and affordable chemicals together with a simple reactor equipment (essentially a stirred tank) are used.

Fenton's discovery of oxidizing radicals enabled the development of other new processes based on this phenomenon, in which various oxidizing radicals, e.g., singlet oxygen, are also generated by physicochemical processes, preferably by the action of UV radiation of various wavelengths, including solar radiation, ultrasound, cold plasma, and streams of electrons, from which dozens can be assembled by combination. This created a huge space for further theoretical and experimental studies of advanced oxidation processes and the verification of their effectiveness and the quality of products for the destruction of hundreds of different organic substances. It should be emphasized that the classic Fenton reaction is an excellent basis for the development of such advanced AOP modifications.

Essentially, AOPs are considered to be reactions where the formation of radicals is initiated by another application of energy intervention, such as ultraviolet radiation. Due to the more powerful generation of hydroxyl radicals, in most cases, AOPs also show a higher efficiency of decomposition of organic substances than the classical Fenton reaction, which is their greatest benefit. Some strongly recalcitrant substances (chloro and nitro derivatives of phenol, phthalates, polyaromatics, polychlorinated aromatics such as PCBs, dioxins and furans, pharmaceuticals, chlorinated fungicides and pesticides, alkyl benzyl sulfonates, etc.) can be significantly disrupted only by these new AOPs. However, they are also more energy intensive, more expensive, and their strong oxidizing power, in such situations where the mineralization of organic substances does not occur, can lead to the formation of oxidation byproducts, which appear more hazardous than the original organic substances. With the classical Fenton reaction, such danger is less serious.

Nevertheless, the simple version of the Fenton reaction is irreplaceable when the concentration of the target pollutant to be removed reaches the values given by regulations

or standards due to its simplicity, incomparably lower costs, and the reduction in the risk of creating hazardous side- and endproducts, which happens frequently. This trend is evident not only from the great interest in studying the application of the Fenton reaction in developing countries, where this simple technology is experiencing a great renaissance, but also in economically developed ones, where it seems to be an ideal decontamination technique for the decontamination of water from various smaller brownfields.

*Why Are These Promising Methods Not Applied in Industry More Commonly?*

AOP is currently receiving extreme attention in literature. AOPs can be made both in the homogeneous variant ($O_3/H_2O_2$, $H_2O_2/Fe^{2+}$ a $H_2O_2/UV$) and the heterogeneous one, e.g., $H_2O_2/TiO_2/UV$. However, the main drawback of the homogeneous system is the inhibition of radical propagation due to the presence of radical scavengers that are commonly present in water, such as carbonates, bicarbonates, and NOM. This is the main reason why homogeneous variants, including the classical Fenton method, are not implemented for the treatment of contaminated waters on an industrial scale, although the application of the Fenton method to various wastewaters is being addressed in laboratories. There have been a large number of similar contributions in the last decade, especially from developing countries, where methods of wastewater treatment from various local sources are beginning to be considered, e.g., from wastewater refineries, the textile industry, especially from dyeworks, tanneries, etc. [14–16]. However, Vega and Valdés [17] documented that the aforementioned limiting factor could be reduced by a heterogeneous arrangement.

## 3. Heterogenous Variant of Fenton Reaction

The simplest variant of the Fenton reaction is the application of metallic Fe particles, including nano-Fe, in which the formation of $Fe^{2+}$ is applied under the oxidation conditions of dissolved oxygen in water and in the presence of metallic iron particles, e.g., as nano-Fe. Although the formation of radicals proceeds through a process similar to that of the homogeneous Fenton reaction, radicals are mainly formed in close proximity to the iron surface, where organic pollutants are also sorbed, which makes it easier for them to meet. Interfering radical absorbers present in the liquid volume have less opportunity to meet them. It is also possible to consider the addition of some heavy metal ions to Fe particles during the oxidation process, which would fulfill the ideal requirements for decontamination reactions based on the Fenton reaction as a universal process, during which organic pollutants would be decontaminated and some heavy metals extracted (sorbed) at the same time. However, how the sorption of heavy metal ions on nanoiron particles could affect the formation of oxidative radicals still remains unknown.

A heterogeneous catalyst is not represented only by particles of metallic Fe. It is also possible to anchor Fe on carriers (e.g., as Fe nanodots on alumina, activated carbon, and various biochars) when the adsorption of the pollutant on the active sites of the sorbent near Fe increases the opportunity for the reaction contact with the catalyst.

It appears that it is possible to incorporate metal ions into the mineralogical grid of clays (montmorillonite, bentonite, natural and synthetic zeolites), especially Fe, Cu, Ce, etc., forming structures known as pillared clays, which exhibit the properties of solid Fenton catalysts, e.g., Fe-ZSM5 [18]. The synthesis of the most common catalyst, Fe-ZSM5, is of great interest, e.g., [19–23], as it is possible to work with them under less acidic conditions. Additionally, they find application in the decontamination of a number of different types of wastewater, e.g., the decolorization of textile waters [24,25], and also pharmaceuticals [13,26,27]. Clays modified in this way act as successful catalysts for the destruction of organic pollutants, e.g., [28,29], (see Figure 1), either as Fenton process catalysts or as photocatalysts or wet oxidation catalysts.

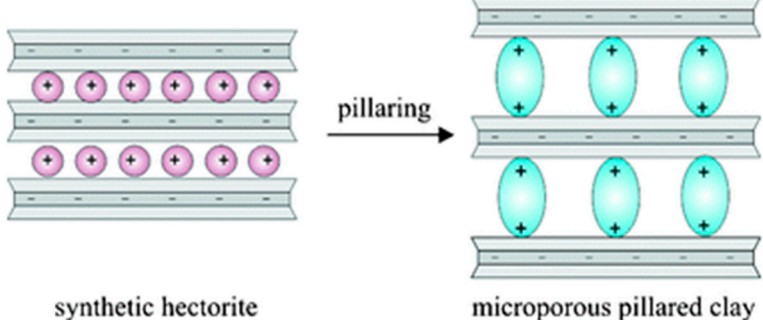

**Figure 1.** Pillaring, exchange of simple cations for polyoxo cations, e.g., Al-Fe polycation $(Al_{13-x}/Fe_x)^{7+}$, see Muñoz et al. [30]. Reproduced with permission from Stöcker et al., realization of truly microporous pillared clays, published by Chemical Communications, 2018 [31].

For this reason, research on heterogeneous catalysts in recent decades has mainly focused on two directions in which it is assumed that the inhibition of oxidative radicals is suppressed. For example, research focuses on catalysts applied in the so-called catalytic wet oxidation condition, which is a process used for the destruction of organic substances using oxygen as an oxidant while the reaction temperatures are characteristically high, usually over 200 °C. A number of solid catalysts have been applied based on various combinations of heavy metals and their oxides, especially transition metals and rare earths, including their various combinations, e.g., Mn/Ce, Co/Bi, Ru/Ce, or CuO-ZnO-Al$_2$O, etc. The reaction mechanism in the three-phase arrangement includes a number of interphase mass transfer steps, and the reaction usually takes place in a bubbled suspension reactor. High temperature and pressure contribute to the formation of oxidizing radical resistant to the possible presence of scavengers. Due to this, it is possible to obtain over 90% conversion of pollutants and mineralization of even complex organic substances into $CO_2$, ammonia and nitrogen, sulfates, phosphates, and chlorides. Wet oxidation and catalytic wet oxidation processes, which enable a significant reduction in temperatures and pressures, have also found industrial implementations for various types of wastewater (pharmaceutical and chemical industry) in the typical temperature range of 140–325 °C and 2–12 MPa. Concerning industrial designs, mainly $Fe^{2+}$/oxygen, $Cu^{2+}$/air or oxygen, Fe-Cu-Mn/$H_2O_2$, and $Fe^{2+}$/air/$H_2O_2$ catalysts were applied. Although the process is expensive and the corrosion of the metal parts of the reactors may cause a problem, it is reported that it can be cheaper than incineration with all the safeguards. All important aspects of this process are excellently described in Bhargava et al. [32]. It can be summarized that even this very promising process can be considered as a continuation of the heterogeneous Fenton reaction realized under high pressures.

Activated carbon, AC/$H_2O_2$ and AC/$O_3$, also plays a very important role. When in the presence of hydrogen peroxide, activated carbon acts as a catalyst because it can produce additional oxidative radicals, and it seems that the inhibition of radicals is suppressed in this system [17]. In the case of ozone, the presence of activated carbon generates oxidizing radicals via the decomposition of the ozone. However, the chemistry of AC surface groups plays a crucial role in radical production. Longer-term contact with hydrogen peroxide disrupts the composition of AC surface functional groups and reduces the formation of radicals due to the increased formation of acidic surface groups. Obviously, the presence of basic groups is mainly responsible for the catalytic decomposition of hydrogen peroxide, which leads to the production of radicals [17]. Nevertheless, other authors favor accelerating the formation of radicals with HNO$_3$ vapors [33]; thus, it would be appropriate to further examine the activated carbon including various biochars as catalysts for the decomposition of hydrogen peroxide. It is evident that not only the selection of suitable ACs and their surface treatment can reveal a longer duration of use in contact with peroxide. Additionally, Vega and Valdés [17] simultaneously showed that the presence of metal ions on the AC surface, mostly Fe, could play a significant role in the decomposition of hydrogen peroxide,

thereby increasing the formation of radicals. Even this variant with AC and hydrogen peroxide can be considered as an $AC/H_2O_2/Fe^{2+}$ application of the Fenton process in a heterogeneous arrangement.

The presence of mesoporous-activated carbon particles is particularly advantageous in waters with a high TDS content (total dissolved soils), which inhibits the formation of radicals during a homogeneous Fenton reaction, as described in the work [34] focused on waters with a high TDS content of around 1%.

It is difficult to decide whether coal acts as a generator of radicals in the presence of dissolved oxygen, or only as a base with active sites for the contact of organic substances with radicals created by other processes than with the contribution of coal, or mainly as a TDS sorbent. Nevertheless, TDS can be sorbed onto the coal and stop interfering with the generation of radicals. Such situations, in the case of waters with a high TDS content (e.g., from tanneries), must be further studied. It is important for the heterogeneous Fenton reaction that the reaction takes place on the surface of the catalyst, and the adsorption of the organic pollutant is one of the important steps for the efficiency of oxidative degradation by radicals (see Figure 2). At different pHs, the isoelectric point of the catalyst would be different, and the electrical charge of the decontaminated target substance would be similarly affected. Simultaneously, the pH of the solution influences the electrostatic relations between the substance and the catalyst (i.e., repel or attract), which can explain the effect of pH on the greater success of the degradation of the target substance under either a homogeneous or heterogeneous Fenton reaction. However, even a pH closer to neutral can save neutralization costs, which are more favorable for heterogeneous Fenton reaction applications. Thus, further research in this direction is still open.

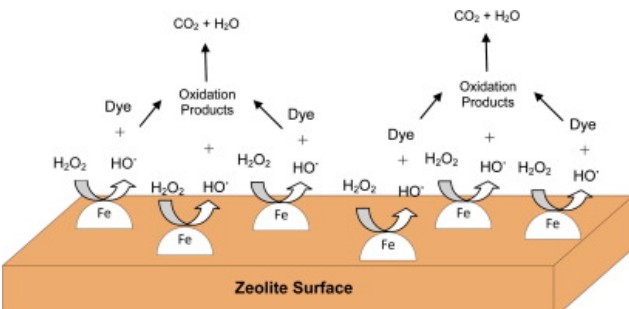

**Figure 2.** Formation of radicals on Fe-zeolite during oxidation of textile dyes. Reproduced with permission from Queirós et al., heterogeneous Fenton reaction oxidation using Fe/ZSM-5 as catalyst in a continuous stirred tank reactor, published by Separation and Purification Technology, 2015 [25].

It is evident that different heterogeneous variants of the Fenton reaction can enlarge AOP for water decontamination at industrially interesting scales and, at reasonable costs, result in useful solutions for environmental issues. For these reasons, considerable attention is currently focused on the development of heterogeneous catalysts with a high and long-term photocatalytic activity for the Fenton and photo-Fenton oxidation of wastewater. A promising heterogeneous variant can work in the future at a neutral pH (which is so far a practically unattainable wish of environmental engineers) and ambient temperature, without the need to neutralize the effluent after treatment. Illustrative examples are particles of magnetite, ($Fe_3O_4$), hematite ($Fe_2O_3$), goethite ($\alpha$-FeOOH), pyrite ($FeS_2$), and lepidocrocite ($\gamma$-FeOOH), although compared to the homogeneous variant, the oxidation reaction rate was usually lower due to the retarding mass transfer effect on the catalyst surface. Hematite and goethite have iron in the $Fe^{3+}$ oxidation state, and for the formation of active radicals, it would be more appropriate to apply UV radiation, which formally follows from the theoretical relationships presented below. It was found that at the same time, they also act as photocatalysts and produce other necessary radicals due to the further reaction of the regenerated $Fe^{2+}$ with peroxide (relation (2) below). Many possible synthetic and natural catalysts have already been described. Essentially, most solid substances

containing transition metals, whether synthetic composites or natural minerals, when in contact with hydrogen peroxide, show catalytic abilities to generate hydroxide oxidation radicals, e.g., [35].

A number of works are devoted to the magnetic surface on powdered activated carbon, which is positive for the practical application, as it is relatively simple to produce a catalyst with magnetite nanoparticles. The coprecipitation reaction (addition of $Fe^{2+}$ and $Fe^{3+}$ salts to treated activated carbon by precipitation with ammonia) is used, see instructions, e.g., [36], while the range of possibilities to degrade various pollutants is really large.

However, the problem with these solid catalysts lies in the gradual leaching of the metal component into the solution and thus in the gradual transition to the conditions of the classic Fenton reaction. The solution to prevent this transition of the heterogeneous Fenton reaction lies in the use of stable nonmetallic catalysts such as graphene [37]. Graphene can act as an effective electrode in the electro-Fenton process [38], as well as graphene oxide, or graphene cornered on $F_3O_4$ particles, which are sacrificed during the reaction [39]. Related to this, it can significantly stabilize and regenerate $Fe^{2+}$ formed by the reaction, i.e., prevent the premature formation of, e.g., $Fe^{3+}$ sludge [40]. It turns out that in such constellation, $Fe^{3+}$ is more active than $Fe^{2+}$; however, carbon nanotubes, diamond nanoparticles [41], or fullerenes [42] could also be used. These carbon species can act either as a separate catalyst or rather as a carrier with a large interfacial area for various combinations of transition metals, e.g., Ce-Fe-RGO (RGO-reduced graphene oxide) [43]. Simultaneously, it has been proved that the Fenton reaction can be operated over a much wider pH range. For example, Wan et al. [43] even stated the pH of 7. Nevertheless, the experimental verification of similarly optimistic data would be necessary.

Current experience with the application of nanocarbon catalysts to the Fenton process is provided by Xin et al. [42]. However, further studies are necessary to verify that the efforts associated with the preparation of these catalysts (e.g., the technique of laser bombardment of graphite, electrospinning of preprepared forms of nanocarbon particles, vapor deposition, etc.) will ensure the increased degradation efficiency, or optimally, the application of this reaction in a pH-neutral environment at reasonable costs. Commercially available activated carbon fibers saturated with $Fe^{3+}$ can act competitively by simple immersion in chloride hexahydrate [44], and reduction, e.g., by borohydride in the case of nanoiron, as stated for the application of the natural fibers of palm biochar [45]. However, activated carbon impregnated with Fe ions can also be used. It was verified by Duarte et al. [46] and Messele et al. [47] for phenol degradation.

In particular, Soares et al. [48] showed that highly mesoporous supports (sol-gel-method-prepared Fe/Xerogel M) and supports with a large interfacial area (Fe/activated carbon M) with 2% Fe content and with increased N- functional groups as catalysts of the Fenton heterogeneous process had greater degradation capabilities than other forms of carbon catalysts in the degradation of p-nitrophenol, 3.6 mmol/L and $H_2O_2$ 29, mmol/L compared to catalysts with carbon nanotubes, which greatly favor the Fenton reaction in a simple version with treated activated carbon (N- functional groups, e.g., after soaking in a solution of melamine or urea). It follows that the surface phenomena, which have not yet been completely clarified, are important at the application of the heterogeneous Fenton reaction. It is obvious that the combination of a high number of mesopores and a high interphase area together with the introduction of suitable functional groups (for *p*-nitrophenol, e.g., N groups) allow for a given wastewater to "tailor a catalyst" for optimal conversions. Therefore, research is also open to easily available activated carbon as a component of the Fenton reaction catalyst.

## 4. When Can the Original Homogeneous Fenton Reaction Be Used?

There are many examples in the literature where real wastewater from various technologies are applied. Unfortunately, the experiments were not carried out on a scale that would demonstrate the possibility of a continuous application of the Fenton reaction, which is essential for industrial use. Most of the published works were applied as demonstra-

tions in glass on a laboratory scale, although experiments were conducted with real water samples collected in effluents from various technologies. Moreover, the Fenton reaction is currently very popular in the photo-Fenton variant, i.e., with the participation of UV radiation, which is another problem for scaling up.

However, the Fenton reaction is cited as the first option for reducing COD values in highly contaminated waters with a high content of dissolved substances, colloids, and NOM, especially strongly colored ones, e.g., for the removal of COD from wastewater after textile dyeing (from dozens of works, e.g., [49]) or even leachate from landfills of various wastes.

The Fenton reaction would be potentially usable as a pretreatment process followed by other processes such as bioremediation, adsorption, membrane separation, sterilization, ion exchange, etc. Another possibility could be its use as a part of a reduction–oxidation node (reduction of chlorinated substances by metal + Fenton reaction). Similarly, it is possible to include (electro)coagulation, filtration, or membrane separation before applying the Fenton process (e.g., Vergili and Gencdal [50], for removing pharmaceuticals). The achievement of a BOD5/COD ratio greater than 0.4 is often considered as a criterion for the suitability of subsequent biodegradation (see, e.g., Cortez et al. [7]). However, the Fenton reaction does not always reach this ratio, and therefore, it is necessary to apply other oxidation processes. If we abandon the use of UV as too expensive, then the combination of $H_2O_2$/ozone can be applied with a small concentration of peroxide, e.g., 400 mg/L [7].

Furthermore, it should be mentioned that reaching the expected final concentrations of the removed substances can be considered a success; however, the decisive factor is whether the resulting water is not ecotoxic after the treatment, i.e., particularly, if it is not carcinogenic and mutagenic. The negative result of the relevant ecotoxicological test, especially, for example, on lower organisms, is completely decisive for expressing the success of the method used. Considering this fact, it is necessary to emphasize the fundamental advantage of the Fenton reaction, namely that even if it usually does not reach the mineralization of dangerous organic substances, it generally does not provide ecotoxic intermediates or products, and related to this, it increases the possibility of the subsequent bioavailability of pollutants.

*Simplified Theoretical Basis of the Fenton Reaction*

Well-known relationships involve the principle of the oxidative radical formation and, simultaneously, the oxidation of the target substance R is indicated. The quantity and quality of oxidation radicals are apparently different. The equations are, for practical reasons, focused on the $\bullet$OH radical, which reveals a very strong oxidizing potential and can be quantified by set methods.

$$Fe^{2+} + H_2O_2 \rightarrow Fe^{3+} + \bullet OH + OH^- \tag{1}$$

$$Fe^{3+} + H_2O_2 \rightarrow Fe^{2+} + HO_2\bullet + H^+ \tag{2}$$

Compared to (1), (2) is several orders of magnitude slower, $Fe^{3+}$ at a pH higher than 5 forms solid oxyhydroxides, and the cycle is completed.

$$RH + \bullet OH \rightarrow R\bullet + H_2O \tag{3}$$

$$R\bullet + H_2O_2 \rightarrow ROH + \bullet OH \tag{4}$$

$$Fe^{3+} + HO\bullet_2 \rightarrow Fe^{2+} + O_2 + H^+ \tag{5}$$

$$R\bullet + O_2 \rightarrow ROO\bullet \tag{6}$$

These radicals, R$\bullet$ and ROO$\bullet$, can further become disproportionate, or produce relatively stable molecules, or react with Fe ions (e.g., [51]). The produced organic intermediates can further react with hydroxyl radicals and with $O_2$ and thus lead to other decomposition products and, in the ideal case, to mineralized components ($H_2O$, $CO_2$, $Cl^-$, etc.), which

is only sometimes completed. A number of similar simplified schemes can be found, as reported in detail, for example, by Umar et al. [52]. However, it is assumed that four main reaction pathways actually occur: the addition of radicals, abstraction of hydrogen, transfer of electrons, and combination of radicals.

However, it is important for practice that the Fenton reaction enables the oxidation and gradual destruction of a number of organic substances, although the exact theoretical background is still not clearly known.

## 5. Problem of the Fenton Process

The biggest problem with the Fenton reaction lies in the fact that the amount of hydrogen peroxide and ferrous ion is not optional and cannot be predicted for real wastewater. Moreover, the decontamination efficiency of various organic pollutants of the target components of interest depends on both the $H_2O_2$:R ratio and the $H_2O_2$:$Fe^{2+}$ ratio. Furthermore, these ratios depend not only on the concentration, chemical composition, and structure of the target substance R but also, in real waters, on a variety of organic and inorganic substances, which are undefined in terms of both their composition and structure. These substances are defined by terms such as natural organic matter (NOM) or dissolved organic matter (DOM), the total amount of dissolved solids, usually salts, total dissolved solids (TDS), chemical oxygen demand (COD, an indicative measure of the amount of oxygen that will be consumed during the reaction), total organic carbon (TOC), or biological oxygen consumption (BOD5), which is the biochemical oxidation of organic or inorganic substances in water under given conditions. The mentioned group quantities are defined by the relevant standardized procedures. Regarding the mechanism of radical formation in AOP, these components consume oxidative radicals; however, they can also produce them under irradiation with light of different wavelengths. Some of them also act as radical absorbers, and heavy metals can even act as catalysts. In addition, these groups show very different values in real waters. For these reasons, to remove the same concentration of the target pollutant (e.g., 100 mg/L benzene) from different wastewaters, a different and difficult method to predict the amount of the substance is needed.

The existence of oxidizing radicals •OH is also very sensitive to both the amount of supplied $H_2O_2$ and $Fe^{2+}$, which can be formally expressed as relationships:

$$Fe^{2+} + \bullet OH \rightarrow Fe^{3+} + OH^- \tag{7}$$

$$H_2O_2 + \bullet OH \rightarrow H_2O + HO_2 \bullet \tag{8}$$

$$HO\bullet_2 + \bullet OH \rightarrow H_2O + O_2 \tag{9}$$

However, the oxidizing power of both $O_2$ itself and other possibly present radicals is much smaller than that of the •OH radical.

It would be theoretically possible to estimate the $H_2O_2$:R ratio from the assumed oxidation schemes for the pure R component for pure water; nevertheless, it is impossible for real water. The ratio must be determined experimentally for each case as evident from data provided by various authors. The values and the ratio of $H_2O_2$:$Fe^{2+}$ for diverse organic substances and real water with different concentrations of the target components R, COD, TOC, and BOD varied considerably, mostly from 2.5–40 mmol/L: 0.17–1.48 mmol/L, and the ratio was usually 20–27. However, the values of this ratio also reached 33 or 3 (at 4 mmol $Fe^{2+}$ for landfill leachate with COD = 743 mg/L, see [7]). For instance, in Hasan et al. [53] and Diya'uddeen et al. [8], the optimal $H_2O_2$ concentration was chosen to be 40.3 mmol/L and for $Fe^{2+}$ was 1.48 mmol/L in a molar ratio of 27.2 for real water containing COD and target phenol in measured concentrations of 2322 mg/L and 84.8 mg/L. However, for high COD values and complicated compositions, e.g., seepage waters containing a number of different micropollutants (phenols, PAHs, pesticides, phthalates, and nonylphenol in microgram quantities), hydrogen peroxide was applied in up to hundreds and thousands of mmol/L in order to achieve a reduction in COD below the recommended 1000 mg/L and such quality of the water, which would make it possible to lead it, for example, to the biological

stage of the WWTP [54]. Even in the work devoted to the treatment of dyeing wastewater containing azo and anthraquinone dyes with a COD over 2500 mg/L, in Gulkaya et al. [55], the authors report a possible supply of 11,323 mmol/L $H_2O_2$ and a ratio of $H_2O_2:Fe^{2+}$ (g/g) up to over 100 for 95% COD removal. Another example of pharmaceutical wastewater (COD = 18,000 mg/L) states 1000 mmol/L $H_2O_2$ and 50 mmol/L $Fe^{2+}$, i.e., the ratio of 20 [50], and a recommendation for hydrogen peroxide dosage in the range of 1–10 mg/L [5], which differs significantly from most designs. It is clear from the published results that there is a significant difference for the estimation of the basic Fenton equation parameters, whether it is the degradation of an organic component in water or in real water, where the synergistic oxidation of all present components, including NOM (usually expressed as COD), takes place. In this case, it is necessary to find the parameters experimentally.

The amount of hydrogen peroxide and the ratio to the catalyst is difficult to estimate even for modifications of the Fenton process. For example, the addition of ozone can reduce the dosage of hydrogen peroxide, while the pH is adjusted towards higher values within which ozone produces radicals. For very complex contaminated leachate from waste dumps (usually with COD above 2000 ng/kg and BOD5 below 100 mg/kg), the applied peroxide concentration was 50 mmol/L and the ratio to $Fe^{2+}$ was 1 [56]. In the case of waters from the production of ammunition containing nitroaromatics and toxic azo components, even for the combination of Fenton with ultrasound, the inverted ratio $Fe^{2+}/H_2O_2 = 500$ (concentration in mg/L) was chosen [57]. Roudi et al. [58] mathematically determined the optimization for landfill leachate values of pH = 3, $Fe^{2+}$ = 781.25 mg/L, $Fe^{2+}/H_2O_2 = 2$, which is in agreement with the value recommended by Cortez et al. [59].

A large number of similar articles can be found in the literature, mostly without a deeper attempt to correlate the obtained data. Closer connections with the content of COD and TOC were sought by Benatti et al. [60], who optimized these amounts for wastewater more generally as the ratios [COD]:[$H_2O_2$] = 1:9 and [$H_2O_2$]:[$Fe^{2+}$] = 4.5:1 (in mg/L, COD in mg/kg).

However, despite a certain chaos in the determination of the basic parameters of the Fenton process, this process has the preconditions to be applied in a real process, certainly as one of the rational pretreatments of heavily contaminated water. For orientation, cost estimates, e.g., to reduce COD minimally by 70% in wastewater from the production of vegetable juices, for [$Fe^{2+}$] = 20 mmol/L, [$H_2O_2$] = 100 mmol/L, pH = 3 a 4 h, the operating procedures were calculated at 4.38 € per $m^3$, see Amaral-Silva et al. [61].

### 5.1. COD and TOC, Important Criteria for Estimating the Inlet $H_2O_2$ Concentration and Ratio $H_2O_2:Fe^{2+}$

Turki et al. [62] searched for the optimum inlet concentration for highly polluted landfill leachate COD of 12,000 mg/L. They found the optimum at an 82 mmol/L $H_2O_2$ and $H_2O_2/Fe^{2+}$ ratio at 4. However, the authors of [63], based on semitheoretical notions of the Fenton mechanism, estimated such quantities as 1.2 mg $H_2O_2$ per 1 mg COD input and 0.9 mg $Fe^{2+}$ per 1 mg COD, which is a smaller molar ratio of the order. Similarly, concerning wastewater from a flax cleaning plant, Abou-Elela et al. [64] indicated a suitable molar ratio of $H_2O_2/Fe^{2+}$ as 25 and a ratio, $H_2O_2:COD$ = 0.75–1, for COD up to 6 g/kg, which is similar to Turki et al. [62]. However, for cork boiling waters with COD = 5000 mg/kg, they reported an optimal ratio of $H_2O_2:COD$ of 2.2 (all in mg/L), an inlet concentration of 311 mmol/L, and an $H_2O_2/Fe^{2+}$ ratio at 8.2.

The theoretical ratio of $H_2O_2:COD$ was originally designed as stoichiometric: 1 g COD = 1 g $O_2$ = 0.03125 mol $O_2$ = 0.0625 mol $H_2O_2$ = 2.125 g $H_2O_2$ [65,66]. It was approximately valid, for example, for small values of COD = 964 mg/L, for the decontamination of wastewater from paper mills with the optimum for $H_2O_2:COD$ = 0.52–1.04 and input values of $H_2O_2$ = 15–30 mmol/L [67]. Similarly, the determination of the most suitable ratio $H_2O_2:COD$ = 2 for the decontamination of synthetic paint (COD = 421, $H_2O_2$ = 800 mg/L, $Fe^{2+}$ = 150 mg/L, molar ratio $H_2O_2:Fe^{2+}$ = 8.8) falls into the specified range [68]. However, these numbers are not generally valid, which is usual for Fenton

reactions in the case of real water. For example, Talebi et al. [69] used 747 mg/L $H_2O_2$ for leachate from landfills with a COD of 3.511 g/L, whereas Saber et al. [70], exploring refinery wastewater, published the ratio of $H_2O_2$:COD = 10.03 and used a COD = 450.

Other authors suggest [53] that rather than relying on COD values, it seems more plausible to consider TOC. They reason that TOC measures carbon that is directly converted and, therefore, is affected neither by the oxidation state of the organic matter nor by organically bound elements such as nitrogen, hydrogen, and inorganic matter. In addition, COD inadequately reflects the actual oxygen requirements for organic pollutant oxidation, as it also includes the oxidation of other substances, e.g., ferrous ions, sulfides, etc. This may explain the reason why the used concentrations of the basic components of the Fenton process and their ratio can be difficult to correlate in real wastewater. However, in the case of a large difference in COD >> TOC concentration, it will be necessary to consider the different chemistry and structure of organic substances, and therefore increase the amount of peroxide. It is very likely that the dosing of chemicals will also affect the BOD and the BOD5:COD ratio and thus also the biodegradability, which opens up space for further research.

### 5.2. Effect of Organic Substances in Effluents (EfOM-Effluent Organic Matter)

The Fenton reaction is often applied to eluents from WWTPs, while effluents from treatment plants contain organic substances, which are, for example, substances resulting from the biological activity of microorganisms, known as so-called extracellular dissolved polymeric substances and soluble microbial products of microbial cell metabolism. These are the ingredients that make up the bulk of COD. Basically, they play the role of sensitizers in photochemical reactions, and therefore, they affect reactions related to UV and solar radiation; moreover, they can also affect the result of an ordinary Fenton reaction, carried out in dark or light environments. Apparently, all advanced oxidation processes are affected by the presence of EfOM, where the main oxidation process takes place with these components, with the micropollutants being degraded as a cometabolic process according to these ideas (see hypothesis in Giannakis et al. [71]). Humic substances absorb light and produce a hydroxyl radical; consequently, further reactions occur with the organic substances and dissolved oxygen present. The entire system, in the presence of hydrogen peroxide and Fe ions, further participates in the formation of oxidative radicals. It is clear that the oxidation process under the conditions of a simple Fenton reaction, but even Fenton under solar radiation (UVA, UVB) or hard UVC radiation, is very complicated, and some opinions of different researchers are mentioned in the publication Giannakis et al. [71].

Obviously, Fenton processes can contain organic sensitizers in different qualities and quantities in different real waters, especially in WWTP effluents; hence, the setting of the most suitable reaction conditions depends mainly on the EfOM content, which significantly varies. The presence of suspended particles would also have an impact, which would affect the penetration of light. Thus, the main process of advanced oxidation by radicals is the destruction of EfOM, while the oxidation of micropollutants is rather accidental and accompanying. Probably, micropollutants would be removed only after EfOM has been removed from these waters. Thus, the preliminary removal of these substances, for example by coagulation or membrane filtration, would be suitable for the deeper decontamination of micropollutants. It can be assumed for Fenton, solar Fenton, and UVB-photoFenton that degradation rates (e.g., pharmaceuticals) are rather slower than with $UVC/H_2O_2$ applications. However, the amount of micropollutants removed by the Fenton reaction can be significant for all reactions.

### 5.3. What Else Needs to Be Clarified concerning the Fenton Reaction?

Considering the fact that TOC and COD are usually associated with contaminated water, it is essential to quantify or realistically estimate the concentration of oxidizing radicals formed by the Fenton reaction necessary for the degradation of organic pollutants in order to achieve the permitted desired target values (concentration of the target component

or final COD, e.g., 1000 mg/kg, zero ecotoxicity, etc.). The amount of oxidizing radicals can be quantified (e.g., [72–75]). Regrettably, the stoichiometric relationship between TOC and the amount of oxidizing radicals is usually not specified. The total amount of generated oxidizing radicals for different concentrations of $H_2O_2$ and $Fe^{2+}$, as well as for their $H_2O_2/Fe^{2+}$ ratio (e.g., for pH 3), is not sufficiently known, which makes a more accurate estimation of the input composition of peroxide and Fe ions for individual cases of contaminated water impossible. Usually, these parameters are selected based on previous experience and are further optimized experimentally. For this reason, it is necessary to accept the situation that the setting of suitable concentrations of basic chemicals and their ratio in the Fenton reaction cannot be reliably determined in advance. Simultaneously, it should be taken into account that the vast majority of works on the Fenton reaction were carried out with simulated water, and their conclusions are unreliable for real water.

## 6. Disadvantages of the Basic Fenton Process and Possible Solutions

The basic disadvantage of the Fenton reaction is the low pH value around pH 3, above which the radicals are not stable. Moreover, at a pH below 3, there is a strong reduction in radical formation [13], and at a pH above 3, specifically around pH 5, insoluble $Fe(OH)_2$ is formed, and at a pH above 4, the low decomposition of peroxide, which preferentially decomposes into oxygen and water without the formation of radicals, appears. For these reasons, it is also necessary to focus on whether the target organic substances are soluble at all of the pH ranges suitable for the generation of radicals. In the case of heterogeneous catalysts, it is also essential to consider the isoelectric point and the surface charge of the catalyst as well as the target substance, i.e., whether the adsorption of the substance on the catalyst is favored at a given pH. It is one of the important control steps of radical oxidation on the catalyst. Treated water also needs to be neutralized before being released into the environment, which increases the cost of chemicals.

It appears that it is possible to apply an additive that "wraps" the Fe ion and prevents the precipitation of iron hydroxide (e.g., by adding resorcinol, see Romero et al. [76]). This direction becomes interesting for further research related to the effort to realize pollutant oxidation at a neutral pH, which is highly urgent.

The application of the Fenton reaction under desired neutral conditions is currently possible only with its modified hybrid version, e.g., with the catalyst ferric-nitrilotriacetate complex ($Fe^{3+}$-NTA) and under the influence of UVA radiation (0.178 mM $Fe^{3+}$-NTA (1:1), 4.54 mM $H_2O_2$, UVA intensity 4.05 mW/cm$^2$, hydraulic retention time (HRT) 2 h, influent pH 7.6 [77]). The application of various chelate complexes with Fe can also be considered as a variant of the homogeneous Fenton reaction, which is also possible in execution with UV. The generation of •OH radicals was also confirmed with the use of $Fe^{3+}$-ethylenediamine-N,N'-disuccinic acid ($Fe^{3+}$-EDDS), and regarding the degradation efficiency, it greatly surpassed other $Fe^{3+}$ complexes, e.g., with citric, malic, oxalic, and wine acid.

Some authors solved the problem of reducing the high consumption of chemicals by treating effluents from treatment plants with extremely low values of iron and peroxide (from popular ratios such as $H_2O_2/Fe^{2+}$ (in mmol/L) = 2/0.2 or 3/0.3. They choose the smallest one, perhaps 1.5/0.1), and the solution was doped with chelates (EDDS and citrate) when it was possible to work at an almost neutral pH, see Miralles-Cuevas et al. [78]. It was verified only for low values of drugs (carbamazepine, flumequine, ibuprofen, ofloxacin, and sulfa-methoxazole) around 15 µg/L and very low values of COD (30–40 mg/L). However, the result of the Fenton reaction, e.g., in the simultaneous presence of complex substances, can be different from the classic Fenton reaction, see Kuznetsova et al. [79], and unlike the classic Fenton reaction, the reaction can also take place at a higher pH, which would be highly favorable. Nevertheless, a more acidic environment would be more favorable even in the presence of chelates and with a heterogeneous Fenton reaction (which is shown by the reduction in •OH oxidation potential formation at a neutral pH (2.8 V at pH 3 vs. 1.9 V at pH 7) [25]).

Additionally, another disadvantage of this reaction is the necessity to subsequently rid the treated water of Fe ions or complexes. Further disadvantages of the homogeneous Fenton process, such as the formation of a large amount of $Fe^{3+}$ and the limited pH range in the very acidic region, prompted the testing of the Fenton process in a heterogeneous form. Additionally, the application in real wastewater encounters the presence of various chelating substances and phosphates, which can react with Fe ions and, thus, interfere with the generation of oxidative radicals, as already mentioned.

## 7. Energy-Assisted Fenton Reactions

As already stated, the simple original Fenton reaction ($Fe^{2+}/H_2O_2$) can be simultaneously combined with UV radiation, ultrasound, or ozonation. Compared to the original classical technology based on simply mixing two common chemicals, such variations are now referred to as "advanced" and are a part of the AOP. The simplest advanced modification of the Fenton reaction is its execution under the application of UV radiation of different wavelengths, which is known as the photo-Fenton variation ($Fe^{2+}/H_2O_2/UV$).

The development of the knowledge of the theoretical foundations of the Fenton process has led to the study and optimization of the $Fe^{2+}$:$H_2O_2$:UV ratio, and logically also to the idea of simplifying the scheme of the photo-Fenton arrangement, for example, by reducing the mass of the catalyst to a minimum, and studying simpler arrangements without the presence of a catalyst. It could even be a system where the amount of catalyst is zero, i.e., the $H_2O_2/UV$ system, in which the oxidizing radicals are generated only by the action of UV radiation on hydrogen peroxide and water. This system is universally applicable to underground and surface wastewater, and it is the only one that enables the large-scale treatment of contaminated water for drinking purposes, perhaps with the exception of perfluorinated compounds. Some successes under strictly laboratory conditions—including photolysis and photo-Fenton for shorter perfluorinated compounds—are cited by Arvaniti and Stasinakis [80].

Photo-Fenton is already an independent process to which dozens of scientific reports are devoted [81–84]. Its simple variation could represent a promising water decontamination process; however, the decontamination products and intermediates may be more toxic than the parent contaminant, and hence there is space for further ecotoxicological research. This danger does not appear with the photocatalysis process. Instead of $Fe^{2+}$ or particles of zero valent Fe in the heterogeneous Fenton variant, other catalysts can be applied, e.g., $TiO_2$, ZnO, i.e., chemical semiconductors which, when irradiated with UV light of certain wavelengths, produce radicals capable of oxidizing organic substances in water [85]. The combination with photocatalysis, when the full use of photogenerated electrons can be expected, which would increase the efficiency of the Fenton process for the application of various semiconductors (such as $TiO_2$, g-$C_3N_4$, graphene, $BiVO_4$, $ZnFeO_4$ and $BiFeO_3$, activated carbon/$CoFe_2O_4$ nanocomposites, etc.) is currently being thoroughly studied [86–88].

From an economic point of view, it would be interesting to replace UV radiation with solar radiation and simultaneously use ultrasounds such as F/US or ozone F/$O_3$. These modifications create different amounts of reactive oxidizing radicals, which are higher than their amount in the classic Fenton process [89]. Variations in the amount of peroxide and the ratio with $Fe^{2+}$ were recorded, e.g., for leachate, 2000 mg/L $H_2O_2$ and 10 mg/L $Fe^{2+}$ were used during aeration under a medium-pressure Hg lamp (125 W). The amount of COD dropped by 57% from 5200 mg/L within 60 min [90]. The decolorization of wastewater especially from the food industry is often preferred. The almost complete decolorization of strongly dark water from olive oil production, which is a significant environmental problem, e.g., for the Mediterranean countries of Europe, was achieved by the combination of $H_2O_2$/Fe (6/1 volumes) under UV radiation [91].

According to the targeted interest of most authors dealing with the oxidative decontamination of organic substances, photo-Fenton, which has been verified in the removal of

various pesticides, antibiotics, or industrial waters [49,92,93] is currently preferred, as well as solar radiation at a neutral pH [94,95] or with heterogeneous catalysts [96,97].

It is evident that the application of heterogeneous variations, including photocatalysis, is more efficient and, due to the reduction in the difficult final sludge, more practical than the homogeneous Fenton reaction. Heterogeneous photo-Fenton with UV radiation could thus be a good alternative for solving a number of water decontamination problems. However, for highly polluted effluents, the penetration of UV radiation through a dense suspension will be strongly reduced. In this case, the solution could be a variation producing UV light using so-called electrodeless lamps with the application of microwaves, which are inserted directly into the catalyst suspension, and the radiation is thus in direct contact with the contaminant, e.g., heterogeneous Fenton + electrodeless lamp UV + microwaves. This combination, which has already been tested for the decontamination of highly polluted seepage waters [98] or without a catalyst for the removal of polybrominated substances by Kastanek et al. [99], could prove useful for point sources of limited volumes of wastewater, e.g., hospitals.

Another modification of the Fenton reaction is the electro-Fenton reaction, in which the two chemicals $H_2O_2$ and $Fe^{2+}$ are produced electrochemically, while the generated radicals have an oxidative effect on the organic pollutants present (see Figure 3). It is an interesting process, suitable for laboratory verification and theoretical research with the potential of practical applications [100–103]. Hydrogen peroxide is formed on the cathode during bubbling with oxygen, which does not need to be added. Simultaneously, the $Fe^{3+}$ ion formed by the Fenton reaction is reduced, which contributes to the renewal of the $Fe^{2+}$ catalyst and the decrease in the ferric ion, thereby reducing the formation of iron sludge, which is one of the inconveniences of the classic Fenton reaction. However, the exact explanations for both radical generation and $Fe^{2+} \rightarrow Fe^{3+} \rightarrow Fe^{2+}$ transitions have not yet been found.

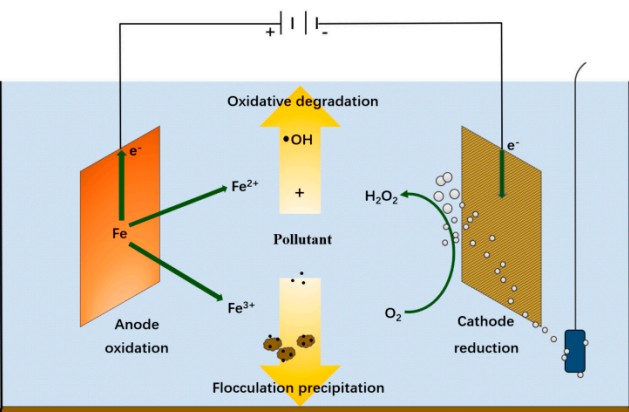

**Figure 3.** Electro-Fenton reaction mechanism diagram [42].

An important role is played by the design, arrangement, and material of the electrodes, especially the cathode, which affects the formation of peroxide [104]). For example, Wang et al. [105] recommends a carbon–polytetrafluoroethylene mixture as a base on a Ni-grid, Zhao et al. [106] recommends graphene, etc. Apparently, the biggest technical problem in practical use is solving the electrode distance for a scale up. Heidari et al. [107] showed that the optimal electrode distance was 3 cm (in a 400 mL beaker, diameter 8.8 cm) for the successful degradation of organics in water, specifically pentachlorophenol. For realistic wastewater flows, it would be necessary to construct a reactor as a multiple system of horizontally and vertically placed pairs of electrodes, which has not yet been tested.

However, the formation of insoluble hydroxide, which is inevitable during the course of the reaction and covers the cathode over time, which reduces the amount of generated hydrogen peroxide, is a problem. It will be difficult to estimate the ratio of peroxide to catalyst. Moreover, regarding real waters, a part of the Fe ions will react with humic

substances and compounds of fulvic acid and NOM to solid flocs, which may dissolve during the reaction. It will be necessary to optimize whether it is more appropriate to separate the flocs (they retain significant amounts of organic substances and possibly heavy metals present) or to leave them in the reaction mixture as a source of catalyst. Coagulation–flocculation is, after all, a common pretreatment method for Fenton reactions, see Vedrenne et al. [108]. An electro-Fenton reaction would generally be suitable for wastewater with a high salt content, owing to the fact that there is no need to dope the water with an electrolyte (especially for the complex composition of, e.g., leachates wastewater from landfills, or wastewater from leather treatment, textile dyeing, etc.). Nevertheless, it is necessary to master the reaction arrangement in real large-volume conditions, namely the material of the electrodes, their size, distances, etc. For example, the so-called reticulated vitreous carbon RVC was applied as a cathode by El-Desoky et al. [109].

Another possibility for the generation of radicals is the use of ultrasound (US), which could be more economically advantageous than UV radiation. It is apparently possible to achieve good efficiencies with less chemical consumption and in a shorter time with the application of US [110]. This is an indisputable advantage that would significantly reduce the amount of sludge with $Fe^{3+}$ [111]. Simultaneously, it is necessary to optimize power USs because increasing the power may cause more formations of cavitation bubbles, which generate oxidative free radicals OH•, and the supersaturation of the bubbles may even cause less implosion, resulting in less efficiency of organic matter destruction.

During the ultrasonic radiation process, there are three reaction degradation zones: hydrophilic substances are localized inside the solution, nonvolatile hydrophobic substances are located mainly at the bubble–water interface, and volatile substances are mostly inside the cavitation bubbles. Pyrolysis, as a degradation reaction, takes plays inside the cavitation bubbles. On the surface of the bubbles, the main reaction is the attack triggered by radicals arising during the implosion of the cavitation bubbles. Last but not least, in the volume of the solution, the reaction takes place with free radicals. Therefore, it is a very complicated process, where each substance is differently hydrophobic, and it is difficult to optimize the entire system [112]. Different ultrasonic variations in the Fenton reaction were again tested, e.g., the electro-Fenton reaction with US, in which hydrogen peroxide and $Fe^{2+}$ were generated in an electrochemical cell with special electrodes (boron diamond doped electrodes) under the influence of US [113], or the completely reduced version without Fe ions (e.g., Rahdar et al. [114]), as an analogy to advanced $UV/H_2O_2$ oxidation. The use of US was also confirmed by Muñoz-Calderon et al. [115], who claimed that the correct dosing of the oxidant and catalyst, supported by ultrasound, could be a good alternative to the Fenton reaction.

## 8. Summarization

The Fenton reaction, by introducing the concept of oxidizing hydroxyl radicals, created the phenomenon of contemporary advanced oxidation processes. It is unique in its apparent simplicity and could significantly contribute to the healing of the environment in all its modifications. The Fenton reaction has been cited and experimentally verified for decades, and although hundreds of publications have been devoted to it, much remains unexplained. Even some very basic data, such as the primary determination of the optimal dosage of two single reaction quantities for the oxidation of the target component in real water, are the subject of rather randomly and/or traditionally chosen quanta, requiring laborious experimental optimizations for each individual case. However, most of the conclusions of a large number of studies on the application of the basic Fenton reaction to different real waters incline to the opinion that basically, regardless of the level of COD and the dosage of peroxide, COD removal efficiencies, even for complexly polluted waters, are on average around 70%. This fact is not only an interesting phenomenon of the Fenton reaction, which does not disappoint in terms of the effectiveness of decontamination even with a certain freedom in the selected amount of basic components, but also a confirmation that it is unique and difficult to be replaced. In addition, it is one of the most important

processes for the remediation of water and soil, especially if the concentrations of pollutants are not high and the cost of the consumed $H_2O_2$ allows for a cost-effective treatment of the reaction medium.

It appears that layered and porous aluminosilicates and analogous double oxides containing iron, copper, or other transition metals exhibit a catalytic activity for the Fenton process. However, as in any catalytic process, the key point is the efficiency of using the reagent, in this case $H_2O_2$, to generate hydroxyl-free radicals. This issue has often been ignored, and mostly only model pollutant extinction and/or total organic carbon reduction has been studied. The optimization of $H_2O_2$ is also considered one of the important parameters in the evaluation of the efficiency of solid catalysts. The current situation in this area does not yet allow conclusions to be drawn regarding the relative catalytic activity of the various tested solids. In addition to the selective activity in $H_2O_2$ conversion, other parameters, such as catalyst stability, absence of leaching, and aging of catalytic sites, as well as operating conditions including pH, temperature, and catalyst amount, must be considered. It will also be necessary to confirm whether nonmetallic catalysts, e.g., graphene, have a high efficiency in generating hydroxyl radicals. Moreover, attention will need to be paid to surface modifications by introducing suitable functional groups on easily available activated carbon, perhaps also in the form of fibers, as a carrier of transit metals, mainly Fe for heterogeneous variations in the Fenton reaction.

Simultaneously, it can be assumed that the use of the Fenton reaction will increase in the near future, motivated by environmental problems and pollution remediation. Future developments in this area will lead to clarification of the current state, with some new materials emerging as major catalysts for the Fenton reaction. It already appears that various modifications of the Fenton method (especially photo-Fenton with UV and visible light) can, in many cases, be more effective for the decomposition of many recalcitrant pollutants than the classical Fenton method. One promising possibility is the verification of the Fenton/ultrasound variation, which can be operated in relatively large volumes, even continuously.

The growing contents of so-called emerging pollutants in waters of all kinds will probably, in the foreseeable future, inevitably lead to the need for their removal even on a real scale, and society will have to bear the necessary costs, which, however, must be rational. Deciding on which advanced oxidation process to use (there will probably be no doubt about its application in the necessary complex with pre- and post-treatment technologies) will be rational, and the choice will probably involve:

- Photolysis of the $UV/H_2O_2$ type, possibly with the application of various catalysts.
- Photocatalysis (most likely $UV/TiO_2$), possibly with the interaction of $H_2O_2$.
- The classic Fenton reaction ($Fe^0$, $Fe^{2+}$, $Fe^{3+}$, possibly in the form of a photo-Fenton reaction under UV exposure).
- Homogeneous Fenton reaction working at neutral pH, i.e., its so-called "green" form corresponding to current environmental trends.

Such a competition is not only still open but also indispensable, and the Fenton process seems to have a great chance. This is evidenced by the incessant publication of its various variants in very good scientific journals.

## 8.1. Future Directions

Research on the Fenton method will continue, and according to the current literature search, its focus should be directed on the following topics:

- Solar energy, including clarifying its real influence on the homogeneous and heterogeneous Fenton reaction, which could be interesting for subtropical and tropical regions, which currently usually suffer from high pollutant loads.
- An evaluation of the possibility of applying other components besides hydrogen peroxide, such as persulphates, percarbonates, or ferrate $Fe^{4+}$, which has been addressed so far by Lee et al. [116] and Karim et al. [117].

- A theoretical clarification of how to optimally dose the amount and ratio of catalyst (preferably $Fe^{2+}$) and $H_2O_2$ for different water compositions, i.e., COD, TOC concentrations, and the composition of various micropollutants, which still, after more than hundreds of years, remains unclear, left either to a random choice or supported by randomized experiments.
- Solving the use of sludge with $Fe^{3+}$ from the point of view of waste policies, especially the use of sludge Fe as a catalyst, which would positively affect the economics of the Fenton process (see Xu et al. [3]).
- Finding $Fe^{3+}$ chelates, which could shift the Fenton reaction to its "green" homogeneous variant, working at a normal pH, which would, among other things, open the Fenton reaction to other possibilities for removing, for example, emerging pollutants from wastewater, when their strong acidification and subsequent neutralization is difficult and expensive, e.g., Lekikot et al. [118], while the removal of the complex from the treated water including regeneration is an indispensable step [119].
- A rational evaluation of costs to achieve the required final values of concentrations and ecotoxicity for individual variants of the Fenton reaction, including hybrid ones, which is necessary for the real use.
- A verification of the Fenton reaction, at least in a pilot setting, on traditional pollutants from selected real producers of these wastes (PAU, BTEX) and on real wastewater from various industries (on paper mill water [67,120], on leachate from landfills [121], as well as emerging pollutants, which is mostly missing). It would make it possible to express a fundamental opinion based on the conviction that, regarding the situation in the issue of wastewater decontamination, the Fenton reaction is an irreplaceable environmental tool.

*8.2. Conclusions*

The more than 100-year-old principle of the Fenton reaction, which fundamentally influenced all subsequent advanced oxidation processes for the removal of pollutants from water and soil, is far from being exhausted. Under real conditions, the formation of oxidative radicals is strongly influenced not only by the presence of natural and dissolved organic substances such as humic and fulvic acids, natural surfactants, and the interaction of soil microorganisms, but also by the presence of natural chelates and other heavy metals. The biggest problem of the Fenton reaction application in the real soil and water environment is the necessity of a low pH of 3, which is ecologically harmful. However, the use of Fe chelates, which enable the reaction at a normal pH, is also not a solution. Various types of substances have been tried, from organic (based on EDTA, amino acids, or peptides) to inorganic (e.g., tripolyphosphates); regrettably, no one has any idea what happens to them in the real environment. The question also arises whether it would be wise to give up the beautiful simplicity of the Fenton reaction and substitute it for complex catalysts, which are more expensive and will eventually be connected with the same problems as simple Fe ions in real waters or soils. As part of the development of scientific knowledge, it is of course necessary to solve problems associated with the development of new catalysts, including nanoforms; nevertheless, it must be taken into consideration that the reality of contaminated soils and waters is complex, and thus experiments in simulated soil and water are rather misleading and will not solve this problem. There is a huge scope for experiments with real soils that would help shed light on the fates of oxidizing radicals under various specific conditions, which could significantly expand scientific knowledge.

In conclusion, it could be stated that there are still many challenging scientific and application issues associated with the Fenton reaction; therefore, we can look forward to the potential upcoming interesting solutions and results related to real applications.

**Author Contributions:** Conceptualization, F.K., O.S. and P.K.; methodology, F.K. and O.S.; validation, F.K., O.S., P.K., M.D. and M.S.; formal analysis, P.K., M.D. and M.S.; investigation, F.K., O.S., P.K., M.D. and M.S.; resources, F.K., O.S. and P.K.; data curation, F.K., O.S. and P.K.; writing—original draft preparation, F.K., O.S., P.K., M.D. and M.S.; writing—review and editing, M.D. and M.S.; visualization,

M.D. and M.S.; supervision, F.K. and O.S.; project administration, O.S.; funding acquisition, O.S. All authors have read and agreed to the published version of the manuscript.

**Funding:** This research was funded by the Ministry of Industry and Trade of the Czech Republic, project no. FV40126. Part of the work was financed by the research infrastructure NanoEnviCZ (LM2018124).

**Data Availability Statement:** Not applicable.

**Acknowledgments:** The financial support of the Ministry of Industry and Trade of the Czech Republic (project no. FV40126) is gratefully acknowledged. Part of the work was financed by the research infrastructure NanoEnviCZ (LM2018124).

**Conflicts of Interest:** The authors declare no conflict of interest.

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
