# Peer review of "Fenton Reaction–Unique but Still Mysterious"

_processes, doi:10.3390/pr11020432_

Round 1

Reviewer 1 Report

In my opinion, the Conclusion section should be rewritten. First of all, it should be shortened. Since the information contained in the current version of this chapter is essential for the entirety of the article, I propose to introduce an additional chapter before the Conclusion chapter. In this new chapter please make a critical discussion and describe  future directions/perspective (based on current chapter named Future Direction). Then enter a short Conclusion chapter.

A few editorial errors are listed below:Line 343: resp. ? Lines 800 - 803: remove bold font; Line 917: please change red colour.

Author Response

Dear Reviewer please see the attachment with responses. Thank you.

Reviewer 2 Report

The review by M. Spacilova et al. considers an important Fenton reaction, which is a key part of advanced oxidation processes used for the degradation of emerging organic pollutants and environment remediation. This work involves an original methodology for the discussion and analysis of contemporary literature data and may be interesting for Processes readers. It can be published after minor revision by addressing the following issues.

 1. The introduction section should be supplemented by highlighting the specific place and aims of this paper among relevant reviews. The methodology of this work should be described in more details.

2. This paper should be supplemented by one-two tables for analytical comparison of processes parameters and catalytic systems along with advantages and drawbacks.

3. The referee suggests that reference list for this review should contain at least ~ 50% of modern literature (5 recent years).

Author Response

(The authors gave the same response as above.)

Reviewer 3 Report

Thank you for giving me the opportunity to revise the MS entitled “Fenton reaction – unique but still mysterious” by Frantisek Kastanek and his/her colleagues that was submitted to “Processes”. The MS submitted is suitable for Processes, and some interesting results were showed. However, there are several requirements that have to consider by the authors. In this regard, the following comments are requested to be addressed by the authors: 

Comment 1: Please modify the keyword.

Comment 2: The novelty of this literature research should be inserted in the text clearly.

Comment 3: I cautiously suggest increasing the relevant content of cost analysis.

Comment 4: The full text needs to be more refined.

Comment 5: Line 362 Please modify the superscript and subscript.

Comment 6: There are many serious formatting and grammar problems in the manuscript. Please rework carefully.

Comment 7: Please carefully check the format of references, such as superscripts, capitalization, journal abbreviations, etc. For example, Line 800 Line 802 Line 956 Line 1010 Line 1043.

I would suggest that the authors review and include the following recent studies to improve the manuscript.

1.    Su, R.; Chai, L.; Tang, C.; Li, B.; Yang, Z., Comparison of the degradation of molecular and ionic ibuprofen in a UV/H2O2 system. Water Sci Technol 2018, 77, (9), 2174-2183.

2.    Su, R.; Dai, X.; Wang, H.; Wang, Z.; Li, Z.; Chen, Y.; Luo, Y.; Ouyang, D., Metronidazole degradation by UV and UV/H2O2 advanced oxidation processes: kinetics, mechanisms, and effects of natural water matrices. Int. J. Environ. Res. Public Health 2022, 19, (19), 12354. 

Best regards,

Author Response

(The authors gave the same response as above.)

Round 2

Reviewer 3 Report

The manuscript has been sufficiently improved to warrant publication in Processes.